# Communication Compression for Decentralized Training

Hanlin Tang[1], Shaoduo Gan[2], Ce Zhang[2], Tong Zhang[3], and Ji Liu[3,1]

[1]Department of Computer Science, University of Rochester
[2]Department of Computer Science, ETH Zurich
[3]Tencent AI Lab
htang14@ur.rochester.edu, sgan@inf.ethz.ch, ce.zhang@inf.ethz.ch,
tongzhang@tongzhang-ml.org, ji.liu.uwisc@gmail.com

## Abstract

*Optimizing distributed learning systems is an art of balancing between computation and communication. There have been two lines of research that try to deal with slower networks:* communication compression *for low bandwidth networks, and* decentralization *for high latency networks. In this paper, We explore a natural question:* can the combination of both techniques lead to a system that is robust to both bandwidth and latency?

*Although the system implication of such combination is trivial, the underlying theoretical principle and algorithm design is challenging: unlike centralized algorithms, simply compressing exchanged information, even in an unbiased stochastic way, within the decentralized network would accumulate the error and fail to converge. In this paper, we develop a framework of compressed, decentralized training and propose two different strategies, which we call* extrapolation compression *and* difference compression*. We analyze both algorithms and prove both converge at the rate of $O(1/\sqrt{nT})$ where $n$ is the number of workers and $T$ is the number of iterations, matching the convergence rate for full precision, centralized training. We validate our algorithms and find that our proposed algorithm outperforms the best of merely decentralized and merely quantized algorithm significantly for networks with both* high latency and low bandwidth.

## 1 Introduction

When training machine learning models in a distributed fashion, the underlying constraints of how workers (or nodes) communication have a significant impact on the training algorithm. When workers cannot form a fully connected communication topology or the communication latency is high (e.g., in sensor networks or mobile networks), decentralizing the communication comes to the rescue. On the other hand, when the amount of data sent through the network is an optimization objective (maybe to lower the cost or energy consumption), or the network bandwidth is low, compressing the traffic, either via sparsification [Wangni et al., 2017, Konečný and Richtárik, 2016] or quantization [Zhang et al., 2017a, Suresh et al., 2017] is a popular strategy. In this paper, our goal is to develop a novel framework that works robustly in an environment that *both* decentralization and communication compression could be beneficial. In this paper, we focus on quantization, the process of lowering the precision of data representation, often in a stochastically unbiased way. But the same techniques would apply to other unbiased compression schemes such as sparsification.

Both decentralized training and quantized (or compressed more generally) training have attracted intensive interests recently [Yuan et al., 2016, Zhao and Song, 2016, Lian et al., 2017a, Konečný and Richtárik, 2016, Alistarh et al., 2017]. Decentralized algorithms usually exchange local models among nodes, which consumes the main communication budget; on the other hand, quantized algorithms usually exchange quantized gradient, and update an un-quantized model. A straightforward idea to combine these two is to directly quantize the models sent through the network during decentralized training. However, this simple strategy does not converge to the right solution as the quantization error would accumulate during training. The technical contribution of this paper is to develop novel algorithms that combine *both* decentralized training and quantized training together.

**Problem Formulation.** We consider the following decentralized optimization:

$$\min_{x \in \mathbb{R}^N} \quad f(x) = \frac{1}{n} \sum_{i=1}^{n} \underbrace{\mathbb{E}_{\xi \sim \mathcal{D}_i} F_i(x; \xi)}_{=:f_i(x)}, \tag{1}$$

where $n$ is the number of node and $\mathcal{D}_i$ is the local data distribution for node $i$. $n$ nodes form a connected graph and each node can only communicate with its neighbors. Here we only assume $f_i(x)$'s are with L-Lipschitzian gradients.

**Summary of Technical Contributions.** In this paper, we propose two decentralized parallel stochastic gradient descent algorithms (D-PSGD): extrapolation compression D-PSGD (ECD-PSGD) and difference compression D-PSGD (DCD-PSGD). Both algorithms can be proven to converge in the rate roughly $O(1/\sqrt{nT})$ where $T$ is the number of iterations. The convergence rates are consistent with two special cases: centralized parallel stochastic gradient descent (C-PSGD) and D-PSGD. To the best of our knowledge, this is the first work to combine quantization algorithms and decentralized algorithms for generic optimization.

The key difference between ECD-PSGD and DCD-PSGD is that DCD-PSGD quantizes the *difference* between the last two local models, and ECD-PSGD quantizes the *extrapolation* between the last two local models. DCD-PSGD admits a slightly better convergence rate than ECD-PSGD when the data variation among nodes is very large. On the other hand, ECD-PSGD is more robust to more aggressive quantization, as extremely low precision quantization can cause DCD-PSGD to diverge, since DCD-PSGD has strict constraint on quantization. In this paper, we analyze both algorithms, and empirically validate our theory. We also show that when the underlying network has both high latency and low bandwidth, both algorithms outperform state-of-the-arts significantly. We present both algorithm because we believe both of them are theoretically interesting. In practice, ECD-PSGD could potentially be a more robust choice.

## 2   Related work

**Stochastic gradient descent** The *Stocahstic Gradient Descent* (**SGD**) [Ghadimi and Lan, 2013, Moulines and Bach, 2011, Nemirovski et al., 2009] - a stochastic variant of the gradient descent method - has been widely used for solving large scale machine learning problems [Bottou, 2010]. It admits the optimal convergence rate $O(1/\sqrt{T})$ for non-convex functions.

**Centralized algorithms** The centralized algorithms is a widely used scheme for parallel computation, such as Tensor-

**Definitions and notations** Throughout this paper, we use following notations and definitions:

- $\nabla f(\cdot)$ denotes the gradient of a function $f$.
- $f^*$ denotes the optimal solution of (1).
- $\lambda_i(\cdot)$ denotes the $i$-th largest eigenvalue of a matrix.
- $\mathbf{1} = [1, 1, \cdots, 1]^\top \in \mathbb{R}^n$ denotes the full-one vector.
- $\| \cdot \|$ denotes the $l_2$ norm for vector.
- $\| \cdot \|_F$ denotes the vector Frobenius norm of matrices.
- $C(\cdot)$ denotes the compressing operator.
- $f_i(x) := \mathbb{E}_{\xi \sim \mathcal{D}_i} F_i(x; \xi)$.

flow [Abadi et al., 2016], MXNet [Chen et al., 2015], and CNTK [Seide and Agarwal, 2016]. It uses a central node to control all leaf nodes. For *Centralized Parallel Stochastic Gradient Descent* (**C-PSGD**), the central node performs parameter updates and leaf nodes compute stochastic gradients based on local information in parallel. In Agarwal and Duchi [2011], Zinkevich et al. [2010], the effectiveness of C-PSGD is studied with latency taken into consideration. The distributed mini-batches SGD, which requires each leaf node to compute the stochastic gradient more than once before the parameter update, is studied in Dekel et al. [2012]. Recht et al. [2011] proposed a variant of C-PSGD, HOGWILD, and proved that it would still work even if we allow the memory to be shared and let the private mode to be overwriten by others. The asynchronous non-convex C-PSGD optimization is studied in Lian et al. [2015]. Zheng et al. [2016] proposed an algorithm to improve the performance of the asynchronous C-PSGD. In Alistarh et al. [2017], De Sa et al. [2017], a quantized SGD is proposed to save the communication cost for both convex and non-convex object functions. The convergence rate for C-PSGD is $O(1/\sqrt{Tn})$. The tradeoff between the mini-batch number and the local SGD step is studied in Lin et al. [2018], Stich [2018].

**Decentralized algorithms** Recently, decentralized training algorithms have attracted significantly amount of attentions. Decentralized algorithms are mostly applied to solve the consensus problem [Zhang et al., 2017b, Lian et al., 2017a, Sirb and Ye, 2016], where the network topology is decentralized. A recent work shows that decentralized algorithms could outperform the centralized counterpart for distributed training [Lian et al., 2017a]. The main advantage of decentralized algorithms over

centralized algorithms lies on avoiding the communication traffic in the central node. In particular, decentralized algorithms could be much more efficient than centralized algorithms when the network bandwidth is small and the latency is large. The decentralized algorithm (also named gossip algorithm in some literature under certain scenarios [Colin et al., 2016]) only assume a connect computational network, without using the central node to collect information from all nodes. Each node owns its local data and can only exchange information with its neighbors. The goal is still to learn a model over all distributed data. The decentralized structure can applied in solving of multi-task multi-agent reinforcement learning [Omidshafiei et al., 2017, Mhamdi et al., 2017]. Boyd et al. [2006] uses a randomized weighted matrix and studied the effectiveness of the weighted matrix in different situations. Two methods [Li et al., 2017, Shi et al., 2015] were proposed to reduce the steady point error in decentralized gradient descent convex optimization. Dobbe et al. [2017] applied an information theoretic framework for decentralize analysis. The performance of the decentralized algorithm is dependent on the second largest eigenvalue of the weighted matrix.

**Decentralized parallel stochastic gradient descent** The *Decentralized Parallel Stochastic Gradient Descent* (**D-PSGD**) [Nedic and Ozdaglar, 2009, Yuan et al., 2016] requires each node to exchange its own stochastic gradient and update the parameter using the information it receives. In Nedic and Ozdaglar [2009], the convergence rate for a time-varying topology was proved when the maximum of the subgradient is assumed to be bounded. In Lan et al. [2017], a decentralized primal-dual type method is proposed with complexity of $O(\sqrt{n/T})$ for general convex objectives. The linear speedup of D-PSGD is proved in Lian et al. [2017a], where the computation complexity is $O(1/\sqrt{nT})$. The asynchronous variant of D-PSGD is studied in Lian et al. [2017b]. In He et al. [2018], they proposed the gradient descent based algorithm (**CoLA**) for decentralized learning of linear classification and regression models, and proved the convergence rate for strongly convex and general convex cases.

**Compression** To guarantee the convergence and correctness, this paper only considers using the unbiased stochastic compression techniques. Existing methods include randomized quantization [Zhang et al., 2017a, Suresh et al., 2017] and randomized sparsification [Wangni et al., 2017, Konečný and Richtárik, 2016]. Other compression methods can be found in Kashyap et al. [2007], Lavaei and Murray [2012], Nedic et al. [2009]. In Drumond et al. [2018], a compressed DNN training algorithm is proposed. In Stich et al. [2018], a centralized biased sparsified parallel SGD with memory is studied and proved to admits an factor of acceleration.

# 3 Preliminary: decentralized parallel stochastic gradient descent (D-PSGD)

Unlike the traditional (centralized) parallel stochastic gradient descent (C-PSGD), which requires a central node to compute the average value of all leaf nodes, the decentralized parallel stochastic gradient descent (D-PSGD) algorithm does not need such a central node. Each node (say node $i$) only exchanges its local model $\boldsymbol{x}^{(i)}$ with its neighbors to take weighted average, specifically, $\boldsymbol{x}^{(i)} = \sum_{j=1}^{n} W_{ij}\boldsymbol{x}^{(j)}$ where $W_{ij} \geq 0$ in general and $W_{ij} = 0$ means that node $i$ and node $j$ is not connected. At $t$th iteration, D-PSGD consists of three steps ($i$ is the node index):

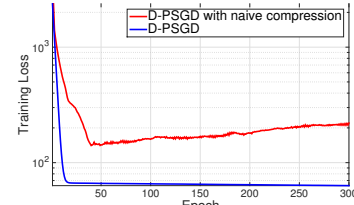

Figure 1: D-PSGD vs. D-PSGD with naive compression

**1.** Each node computes the stochastic gradient $\nabla F_i(\boldsymbol{x}_t^{(i)}; \xi_t^{(i)})$, where $\xi_t^{(i)}$ is the samples from its local data set and $\boldsymbol{x}_t^{(i)}$ is the local model on node $i$.

**2.** Each node queries its neighbors' variables and updates its local model using $\boldsymbol{x}^{(i)} = \sum_{j=1}^{n} W_{ij}\boldsymbol{x}^{(j)}$.

**3.** Each node updates its local model $\boldsymbol{x}_t^{(i)} \leftarrow \boldsymbol{x}_t^{(i)} - \gamma_t \nabla F_i\left(\boldsymbol{x}_t^{(i)}; \xi_t^{(i)}\right)$ using stochastic gradient, where $\gamma_t$ is the learning rate.

To look at the D-PSGD algorithm from a global view, by defining

$$X := [\boldsymbol{x}^{(1)}, \boldsymbol{x}^{(2)}, \cdots, \boldsymbol{x}^{(n)}] \in \mathbb{R}^{N \times n}, \quad G(X; \xi) := [\nabla F_1(x^{(1)}; \xi^{(1)}), \cdots, \nabla F_n(x^{(n)}; \xi^{(n)})]$$

$$\nabla f(\overline{X}) := \sum_{i=1}^{n} \frac{1}{n} \nabla f_i\left(\frac{1}{n} \sum_{i=1}^{n} x^{(i)}\right), \quad \overline{\nabla f}(X) := \mathbb{E}_\xi G(X; \xi_t)\frac{\mathbf{1}}{n} = \frac{1}{n} \sum_{i=1}^{n} \nabla f_i(x^{(i)}),$$

the D-PSGD can be summarized into the form $X_{t+1} = X_t W - \gamma_t G(X_t; \xi_t)$.

The convergence rate of D-PSGD can be shown to be $O\left(\frac{\sigma}{\sqrt{nT}} + \frac{n^{\frac{1}{3}}\zeta^{\frac{2}{3}}}{T^{\frac{2}{3}}}\right)$ (without assuming convexity) where both $\sigma$ and $\zeta$ are the stochastic variance (please refer to Assumption 1 for detailed definitions), if the learning rate is chosen appropriately.

# 4 Quantized, Decentralized Algorithms

We introduce two quantized decentralized algorithms that compress information exchanged between nodes. All communications for decentralized algorithms are exchanging local models $\boldsymbol{x}^{(i)}$.

To reduce the communication cost, a straightforward idea is to compress the information exchanged within the decentralized network just like centralized algorithms sending compressed stochastic gradient [Alistarh et al., 2017]. Unfortunately, such naive combination does not work even using the unbiased stochastic compression and diminishing learning rate as shown in Figure 1. The reason can be seen from the detailed derivation (please find it in Supplement).

Before propose our solutions to this issue, let us first make some common optimization assumptions for analyzing decentralized stochastic algorithms [Lian et al., 2017b].

**Assumption 1.** *Throughout this paper, we make the following commonly used assumptions:*

1. ***Lipschitzian gradient:*** *All function $f_i(\cdot)$'s are with L-Lipschitzian gradients.*
2. ***Symmetric double stochastic matrix:*** *The weighted matrix $W$ is a real double stochastic matrix that satisfies $W = W^\top$ and $W\mathbf{1} = W$.*
3. ***Spectral gap:*** *Given the symmetric doubly stochastic matrix $W$, we define $\rho := \max\{|\lambda_2(W)|, |\lambda_n(W)|\}$ and assume $\rho < 1$.*
4. ***Bounded variance:*** *Assume the variance of stochastic gradient to be bounded*

$$\mathbb{E}_{\xi \sim \mathcal{D}_i} \|\nabla F_i(\boldsymbol{x}; \xi) - \nabla f_i(\boldsymbol{x})\|^2 \leqslant \sigma^2, \quad \frac{1}{n}\sum_{i=1}^n \|\nabla f_i(\boldsymbol{x}) - \nabla f(\boldsymbol{x})\|^2 \leqslant \zeta^2, \quad \forall i, \forall \boldsymbol{x},$$

5. ***Start from 0:*** *We assume $X_1 = 0$. This assumption simplifies the proof w.l.o.g.*
6. ***Independent and unbiased stochastic compression:*** *The stochastic compression operation $\boldsymbol{C}(\cdot)$ is unbiased, that is, $\mathbb{E}(\boldsymbol{C}(Z)) = Z$ for any $Z$ and the stochastic compressions are independent on different workers or at different time point.*

The last assumption essentially restricts the compression to be lossy but unbiased. Biased stochastic compression is generally hard to ensure the convergence and lossless compression can combine with any algorithms. Both of them are beyond of the scope of this paper. The commonly used stochastic unbiased compression include random quantization[1] [Zhang et al., 2017a] and sparsification[2] [Wangni et al., 2017, Konečný and Richtárik, 2016].

## 4.1 Difference compression approach

In this section, we introduces a difference based approach, namely, difference compression D-PSGD (DCD-PSGD), to ensure efficient convergence.

The DCD-PSGD basically follows the framework of D-PSGD, except that nodes exchange the compressed difference of local models between two successive iterations, instead of exchanging local models. More specifically, each node needs to store its neighbors' models in last iteration $\{\hat{\boldsymbol{x}}_t^{(j)} : j$ is node $i$'s neighbor$\}$ and follow the following steps:

1. take the weighted average and apply stochastic gradient descent step: $\boldsymbol{x}_{t+\frac{1}{2}}^{(i)} = \sum_{j=1}^n W_{ij}\hat{\boldsymbol{x}}_t^{(j)} - \gamma\nabla F_i(\boldsymbol{x}_t^{(i)}; \xi_t^{(i)})$, where $\hat{\boldsymbol{x}}_t^{(j)}$ is just the replica of $\boldsymbol{x}_t^{(j)}$ but is stored on node $i$[3];
2. compress the difference between $\boldsymbol{x}_t^{(i)}$ and $\boldsymbol{x}_{t+\frac{1}{2}}^{(i)}$ and update the local model: $\boldsymbol{z}_t^{(i)} = \boldsymbol{x}_{t+\frac{1}{2}}^{(i)} - \boldsymbol{x}_t^{(i)}, \quad \boldsymbol{x}_{t+1}^{(i)} = \boldsymbol{x}_t^{(i)} + \boldsymbol{C}(\boldsymbol{z}_t^{(i)})$;
3. send $\boldsymbol{C}(\boldsymbol{z}_t^{(i)})$ and query neighbors' $\boldsymbol{C}(\boldsymbol{z}_t)$ to update the local replica: $\forall j$ is node $i$'s neighbor, $\hat{\boldsymbol{x}}_{t+1}^{(j)} = \hat{\boldsymbol{x}}_t^{(j)} + \boldsymbol{C}(\boldsymbol{z}_t^{(j)})$.

The full DCD-PSGD algorithm is described in Algorithm 1.

To ensure convergence, we need to make some restriction on the compression operator $\boldsymbol{C}(\cdot)$. Again this compression operator could be random quantization or random sparsification or any

**Algorithm 1** DCD-PSGD

1: **Input:** Initial point $\boldsymbol{x}_1^{(i)} = \boldsymbol{x}_1$, initial replica $\hat{\boldsymbol{x}}_1^{(i)} = \boldsymbol{x}_1$, iteration step length $\gamma$, weighted matrix $W$, and number of total iterations T
2: **for** t = 1,2,...,T **do**
3:    Randomly sample $\xi_t^{(i)}$ from local data of the $i$th node
4:    Compute local stochastic gradient $\nabla F_i(\boldsymbol{x}_t^{(i)}; \xi_t^{(i)})$ using $\xi_t^{(i)}$ and current optimization variable $\boldsymbol{x}_t^{(i)}$
5:    Update the local model using local stochastic gradient and the weighted average of its connected neighbors' replica (denote as $\hat{\boldsymbol{x}}_t^{(j)}$):
$$\boldsymbol{x}_{t+\frac{1}{2}}^{(i)} = \sum_{j=1}^{n} W_{ij}\boldsymbol{x}_t^{(j)} - \gamma\nabla F_i(\boldsymbol{x}_t^{(i)}; \xi_t^{(i)}),$$
6:    Each node computes $\boldsymbol{z}_t^{(i)} = \boldsymbol{x}_{t+\frac{1}{2}}^{(i)} - \boldsymbol{x}_t^{(i)}$, and compress this $\boldsymbol{z}_t^{(i)}$ into $\boldsymbol{C}(\boldsymbol{z}_t^{(i)})$.
7:    Update the local optimization variables
$$\boldsymbol{x}_{t+1}^{(i)} \leftarrow \boldsymbol{x}_t^{(i)} + \boldsymbol{C}(\boldsymbol{z}_t^{(i)}).$$
8:    Send $\boldsymbol{C}(\boldsymbol{z}_t^{(i)})$ to its connected neighbors, and update the replicas of its connected neighbors' values:
$$\hat{\boldsymbol{x}}_{t+1}^{(j)} = \hat{\boldsymbol{x}}_t^{(j)} + \boldsymbol{C}(\boldsymbol{z}_t^{(i)}).$$
9: **end for**
10: **Output:** $\frac{1}{n}\sum_{i=1}^{n}\boldsymbol{x}_T^{(i)}$

**Algorithm 2** ECD-PSGD

1: **Input:** Initial point $\boldsymbol{x}_1^{(i)} = \boldsymbol{x}_1$, initial estimate $\tilde{\boldsymbol{x}}_1^{(i)} = \boldsymbol{x}_1$, iteration step length $\gamma$, weighted matrix $W$, and number of total iterations T.
2: **for** $t = 1, 2, \cdots, T$ **do**
3:    Randomly sample $\xi_t^{(i)}$ from local data of the $i$th node
4:    Compute local stochastic gradient $\nabla F_i(\boldsymbol{x}_t^{(i)}; \xi_t^{(i)})$ using $\xi_t^{(i)}$ and current optimization variable $\boldsymbol{x}_t^{(i)}$
5:    Compute the neighborhood weighted average by using the estimate value of the connected neighbors :
$$\boldsymbol{x}_{t+\frac{1}{2}}^{(i)} = \sum_{j=1}^{n} W_{ij}\tilde{\boldsymbol{x}}_t^{(j)}$$
6:    Update the local model
$$\boldsymbol{x}_{t+1}^{(i)} \leftarrow \boldsymbol{x}_{t+\frac{1}{2}}^{(i)} - \gamma\nabla F_i(\boldsymbol{x}_t^{(i)}, \xi_t^{(i)})$$
7:    Each node computes the $z$-value of itself:
$$\boldsymbol{z}_{t+1}^{(i)} = (1 - 0.5t)\boldsymbol{x}_t^{(i)} + 0.5t\boldsymbol{x}_{t+1}^{(i)}$$
and compress this $\boldsymbol{z}_t^{(i)}$ into $\boldsymbol{C}(\boldsymbol{z}_t^{(j)})$.
8:    Each node updates the estimate for its connected neighbors:
$$\tilde{\boldsymbol{x}}_{t+1}^{(j)} = \left(1 - 2t^{-1}\right)\tilde{\boldsymbol{x}}_t^{(j)} + 2t^{-1}\boldsymbol{C}(\boldsymbol{z}_t^{(j)})$$
9: **end for**
10: **Output:** $\frac{1}{n}\sum_{i=1}^{n}\boldsymbol{x}_T^{(i)}$

---

other operators. We introduce the definition of the signal-to-noise related parameter $\alpha$. Let $\alpha := \sqrt{\sup_{Z\neq 0}\|Q\|_F^2/\|Z\|_F^2}$, where $Q = Z - C(Z)$. We have the following theorem.

**Theorem 1.** *Let $\mu := \max_{i\in\{2,\cdots,n\}}|\lambda_i - 1|$. If $\alpha$ satisfies $(1-\rho)^2 - 4\mu^2\alpha^2 > 0$ and $\gamma$ satisfies $1 - 3D_1L^2\gamma^2 > 0$, then under the Assumption 1, , we have the following convergence rate for Algorithm 1:*

$$\sum_{t=1}^{T}\left((1 - D_3)\,\mathbb{E}\|\nabla f(\overline{X}_t)\|^2 + D_4\mathbb{E}\|\overline{\nabla f}(X_t)\|^2\right) \leq \frac{2(f(0) - f^*)}{\gamma} + \frac{L\gamma T\sigma^2}{n}$$

$$+ \left(\frac{T\gamma^2 LD_2}{2} + \frac{\left(4L^2 + 3L^3D_2\gamma^2\right)D_1T\gamma^2}{1 - 3D_1L^2\gamma^2}\right)\sigma^2 + \frac{\left(4L^2 + 3L^3D_2\gamma^2\right)3D_1T\gamma^2}{1 - 3D_1L^2\gamma^2}\zeta^2, \qquad (2)$$

*where*

$$D_1 := \frac{2\alpha^2}{1 - \rho^2}\left(\frac{2\mu^2(1 + 2\alpha^2)}{(1-\rho)^2 - 4\mu^2\alpha^2} + 1\right) + \frac{1}{(1-\rho)^2}, \quad D_2 := 2\alpha^2\left(\frac{2\mu^2(1 + 2\alpha^2)}{(1-\rho)^2 - 4\mu^2\alpha^2} + 1\right)$$

$$D_3 := \frac{\left(4L^2 + 3L^3D_2\gamma^2\right)3D_1\gamma^2}{1 - 3D_1L^2\gamma^2} + \frac{3LD_2\gamma^2}{2}, \qquad D_4 := (1 - L\gamma).$$

To make the result more clear, we appropriately choose the steplength in the following:

**Corollary 2.** *Let $D_1$, $D_2$, $\mu$ follow to same definition in Theorem 1, and choose $\gamma = \left(6\sqrt{D_1}L + 6\sqrt{D_2}L + \frac{\sigma}{\sqrt{n}}T^{\frac{1}{2}} + \zeta^{\frac{2}{3}}T^{\frac{1}{3}}\right)^{-1}$ in Algorithm 1. If $\alpha$ is small enough that satisfies $(1-\rho)^2 - 4\mu^2\alpha^2 > 0$, then we have*

$$\frac{1}{T}\sum_{t=1}^{T}\mathbb{E}\|\nabla f(\overline{X}_t)\|^2 \lesssim \frac{\sigma}{\sqrt{nT}} + \frac{\zeta^{\frac{2}{3}}}{T^{\frac{2}{3}}} + \frac{1}{T},$$

*if we treat $f(0) - f^*$, $L$, and $\rho$ constants.*

The leading term of the convergence rate is $O\left(1/\sqrt{Tn}\right)$, and we also proved the convergence rate for $\mathbb{E}\left[\sum_{i=1}^{n}\left\|\overline{X}_t - \boldsymbol{x}_t^{(i)}\right\|^2\right]$ (see (27) in Supplementary). We shall see the tightness of our result in the following discussion.

**Linear speedup** Since the leading term of the convergence rate is $O\left(1/\sqrt{Tn}\right)$ when $T$ is large, which is consistent with the convergence rate of C-PSGD, this indicates that we would achieve a linear speed up with respect to the number of nodes.

**Consistence with D-PSGD** Setting $\alpha = 0$ to match the scenario of D-PSGD, ECD-PSGD admits the rate $O\left(\frac{\sigma}{\sqrt{nT}} + \frac{\zeta^{\frac{2}{3}}}{T^{\frac{2}{3}}}\right)$, that is slightly better the rate of D-PSGD proved in Lian et al. [2017b] $O\left(\frac{\sigma}{\sqrt{nT}} + \frac{n^{\frac{2}{3}}\zeta^{\frac{2}{3}}}{T^{\frac{2}{3}}}\right)$. The non-leading terms' dependence of the spectral gap $(1 - \rho)$ is also consistent with the result in D-PSDG.

## 4.2 Extrapolation compression approach

From Theorem 1, we can see that there is an upper bound for the compressing level $\alpha$ in DCD-PSGD. Moreover, since the spectral gap $(1 - \rho)$ would decrease with the growth of the amount of the workers, so DCD-PSGD will fail to work under a very aggressive compression. So in this section, we propose another approach, namely ECD-PSGD, to remove the restriction of the compressing degree, with a little sacrifice on the computation efficiency.

For ECD-PSGD, we make the following assumption that the noise brought by compression is bounded.

**Assumption 2.** (**Bounded compression noise**) *We assume the noise due to compression is unbiased and its variance is bounded, that is, $\forall z \in \mathbb{R}^n$*
$$\mathbb{E}\|C(z) - z\|^2 \leq \tilde{\sigma}^2/2, \quad \forall z$$
Instead of sending the local model $x_t^{(i)}$ directly to neighbors, we send a $z$-value that is extrapolated from $x_t^{(i)}$ and $x_{t-1}^{(i)}$ at each iteration. Each node (say, node $i$) estimates its neighbor's values $x_t^{(j)}$ from compressed $z$-value at $t$-th iteration. This procedure could ensure diminishing estimate error, in particular, $\mathbb{E}\|\tilde{x}_t^{(j)} - x_t^{(j)}\|^2 \leq \mathcal{O}(t^{-1})$. To satisfy Assumption 2, one may need to use the quantization strategy sensitive to the magnitude of $z$. But our experiments show that using fix precision randomized quantization strategy - a magnitude independent method - still works very well.

At $t$th iteration, node $i$ performs the following steps to estimate $x_t^{(j)}$ by $\tilde{x}_t^{(j)}$:

- The node $j$, computes the $z$-value that is obtained through extrapolation
$$z_t^{(j)} = (1 - 0.5t)\, x_{t-1}^{(j)} + 0.5t x_t^{(j)}, \tag{3}$$

- Compress $z_t^{(j)}$ and send it to its neighbors, say node $i$. Node $i$ computes $\tilde{x}_t^{(j)}$ using
$$\tilde{x}_t^{(j)} = \left(1 - 2t^{-1}\right) \tilde{x}_{t-1}^{(j)} + 2t^{-1} C(z_t^{(j)}). \tag{4}$$

Using Lemma 12 (see Supplemental Materials), if the compression noise $q_t^{(j)} := z_t^{(i)} - C(z_t^{(i)})$ is globally bounded variance by $\tilde{\sigma}^2/2$, we will have
$$\mathbb{E}(\|\tilde{x}_t^{(j)} - x_t^{(j)}\|^2) \leq \tilde{\sigma}^2/t.$$
Using this way to estimate the neighbors' local models leads to the following equivalent updating form
$$X_{t+1} = \tilde{X}_t W - \gamma_t G(X_t; \xi_t) = X_t W + \underbrace{Q_t W}_{\text{diminished estimate error}} - \gamma_t G(X_t; \xi_t).$$

The full extrapolation compression D-PSGD (ECD-PSGD) algorithm is summarized in Algorithm 2.

Below we will show that EDC-PSGD algorithm would admit the same convergence rate and the same computation complexity as D-PSGD.

**Theorem 3** (Convergence of Algorithm 2). *Under Assumptions 1 and 2, choosing $\gamma_t$ in Algorithm 2 to be constant $\gamma$ satisfying $1 - 6C_1 L^2 \gamma^2 > 0$, we have the following convergence rate for Algorithm 2*
$$\sum_{t=1}^{T} \left((1 - C_3)\, \mathbb{E}\|\nabla f(\overline{X}_t)\|^2 + C_4 \mathbb{E}\|\overline{\nabla f}(X_t)\|^2\right)$$
$$\leq \frac{2(f(0) - f^*)}{\gamma} + \frac{L \log T}{n\gamma}\tilde{\sigma}^2 + \frac{LT\gamma}{n}\sigma^2 + \frac{4C_2 \tilde{\sigma}^2 L^2}{1 - \rho^2} \log T + 4L^2 C_2 \left(\sigma^2 + 3\zeta^2\right) C_1 T \gamma^2, \tag{5}$$
*where $C_1 := \frac{1}{(1-\rho)^2}$, $C_2 := \frac{1}{1 - 6\rho^{-2} C_1 L^2 \gamma^2}$, $C_3 := 12L^2 C_2 C_1 \gamma^2$, and $C_4 := 1 - L\gamma$.*

To make the result more clear, we choose the steplength in the following:

**Corollary 4.** *If choosing the steplength $\gamma = \left(12\sqrt{C_1}L + \frac{\sigma}{\sqrt{n}}T^{\frac{1}{2}} + \zeta^{\frac{2}{3}}T^{\frac{1}{3}}\right)^{-1}$, then Algorithm 2 admits the following convergence rate (with $f(0) - f^*$, $L$, and $\rho$ treated as constants):*
$$\frac{1}{T}\sum_{t=1}^{T} \mathbb{E}\|\nabla f(\overline{X}_t)\|^2 \lesssim \frac{\sigma(1 + \frac{\tilde{\sigma}^2 \log T}{n})}{\sqrt{nT}} + \frac{\zeta^{\frac{2}{3}}(1 + \frac{\tilde{\sigma}^2 \log T}{n})}{T^{\frac{2}{3}}} + \frac{1}{T} + \frac{\tilde{\sigma}^2 \log T}{T}. \tag{6}$$

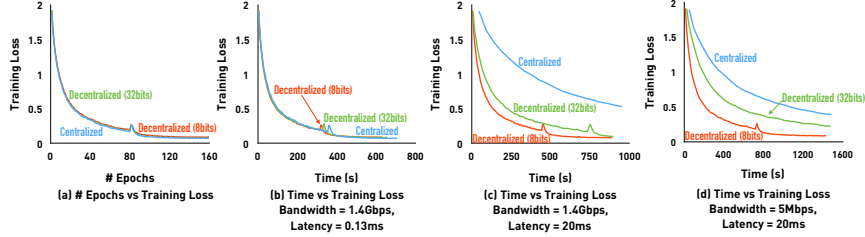

Figure 2: Performance Comparison between Decentralized and AllReduce implementations.

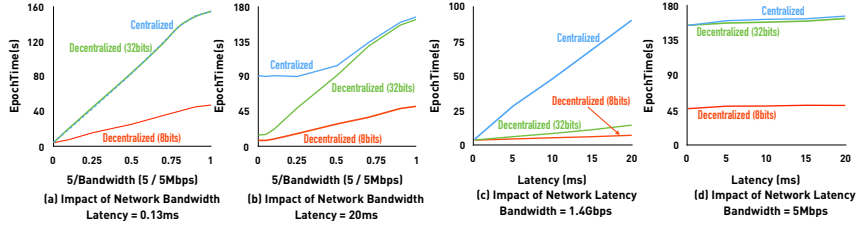

Figure 3: Performance Comparison in Diverse Network Conditions.

This result suggests the algorithm converges roughly in the rate $O(1/\sqrt{nT})$, and we also proved the convergence rate for $\mathbb{E}\left[\sum_{i=1}^{n}\left\|\overline{X}_t - \boldsymbol{x}_t^{(i)}\right\|^2\right]$ (see (36) in Supplementary). The followed analysis will bring more detailed interpretation to show the tightness of our result.

**Linear speedup** Since the leading term of the convergence rate is $O(1/\sqrt{nT})$ when $T$ is large, which is consistent with the convergence rate of C-PSGD, this indicates that we would achieve a linear speed up with respect to the number of nodes.

**Consistence with D-PSGD** Setting $\tilde{\sigma} = 0$ to match the scenario of D-PSGD, ECD-PSGD admits the rate $O\left(\frac{\sigma}{\sqrt{nT}} + \frac{\zeta^{\frac{2}{3}}}{T^{\frac{2}{3}}}\right)$, that is slightly better the rate of D-PSGD proved in Lian et al. [2017b] $O\left(\frac{\sigma}{\sqrt{nT}} + \frac{n^{\frac{1}{3}}\zeta^{\frac{2}{3}}}{T^{\frac{2}{3}}}\right)$. The non-leading terms' dependence of the spectral gap $(1-\rho)$ is also consistent with the result in D-PSDG.

**Comparison between DCD-PSGD and ECD-PSGD** On one side, in term of the convergence rate, ECD-PSGD is slightly worse than DCD-PSGD due to additional terms $\left(\frac{\sigma\tilde{\sigma}^2 \log T}{n\sqrt{nT}} + \frac{\zeta^{\frac{2}{3}}\tilde{\sigma}^2 \log T}{nT^{\frac{2}{3}}} + \frac{\tilde{\sigma}^2 \log T}{T}\right)$ that suggests that if $\tilde{\sigma}$ is relatively large than $\sigma$, the additional terms dominate the convergence rate. On the other side, DCD-PSGD does not allow too aggressive compression or quantization and may lead to diverge due to $\alpha \le \frac{1-\rho}{2\sqrt{2}\mu}$, while ECD-PSGD is quite robust to aggressive compression or quantization.

## 5 Experiments

In this section we evaluate two decentralized algorithms by comparing with an allreduce implementation of centralized SGD. We run experiments under diverse network conditions and show that, decentralized algorithms with low precision can speed up training without hurting convergence.

### 5.1 Experimental Setup

We choose the image classification task as a benchmark to evaluate our theory. We train ResNet-20 [He et al., 2016] on CIFAR-10 dataset which has 50,000 images for training and 10,000 images for testing. Two proposed algorithms are implemented in Microsoft CNTK and compablack with CNTK's original implementation of distributed SGD:

• **Centralized:** This implementation is based on MPI allreduce primitive with full precision (32 bits). It is the standard training method for multiple nodes in CNTT.

• **Decentralized_32bits/8bits:** The implementation of the proposed decentralized approach with OpenMPI. The full precision is 32 bits, and the compressed precision is 8 bits.

• In this paper, we omit the comparison with quantized centralized training because the difference between Decentralized 8bits and Centralized 8bits would be similar to the original decentralized training paper Lian et al. [2017a] – when the network latency is high, decentralized algorithm outperforms centralized algorithm in terms of the time for each epoch.

We run all experiments on 8 Amazon $p2.xlarge$ EC2 instances, each of which has one Nvidia K80 GPU. We use each GPU as a node. In decentralized cases, 8 nodes are connected in a ring topology, which means each node just communicates with its two neighbors. The batch size for each node is same as the default configuration in CNTK. We also tune learning rate for each variant.

## 5.2 Convergence and Run Time Performance

We first study the convergence of our algorithms. Figure 2(a) shows the convergence w.r.t # epochs of centralized and decentralized cases. We only show ECD-PSGD in the figure (and call it Decentralized) because DCD-PSGD has almost identical convergence behavior in this experiment. We can see that with our algorithms, decentralization and compression would not hurt the convergence rate.

We then compare the runtime performance. Figure 2(b, c, d) demonstrates how training loss decreases with the run time under different network conditions. We use $tc$ command to change bandwidth and latency of the underlying network. By default, 1.4 Gbps bandwidth and 0.13 ms latency is the best network condition we can get in this cluster. On this occasion, all implementations have a very similar runtime performance because communication is not the bottleneck for system. When the latency is high, however, decentralized algorithms outperform the allreduce because of fewer number of communications. In comparison with decentralized full precision cases, low precision methods exchange much less amount of data and thus can outperform full precision cases in low bandwidth situation, as is shown in Figure 2(d).

## 5.3 Speedup in Diverse Network Conditions

To better understand the influence of bandwidth and latency on speedup, we compare the time of one epoch under various of network conditions. Figure 3(a, b) shows the trend of epoch time with bandwidth decreasing from 1.4 Gbps to 5 Mbps. When the latency is low (Figure 3(a)), low precision algorithm is faster than its full precision counterpart because it only needs to exchange around one fourth of full precision method's data amount. Note that although in a decentralized way, full precision case has no advantage over allreduce in this situation, because they exchange exactly the same amount of data. When it comes to high latency shown in Figure 3(b), both full and low precision cases are much better than allreduce in the beginning. But also, full precision method gets worse dramatically with the decline of bandwidth.

Figure 3(c, d) shows how latency influences the epoch time under good and bad bandwidth conditions. When bandwidth is not the bottleneck (Figure 3(c)), decentralized approaches with both full and low precision have similar epoch time because they have same number of communications. As is expected, allreduce is slower in this case. When bandwidth is very low (Figure 3(d)), only decentralized algorithm with low precision can achieve best performance among all implementations.

## 5.4 Discussion

Our previous experiments validate the efficiency of the decentralized algorithms on 8 nodes with 8 bits. However, we wonder if we can scale it to more nodes or compress the exchanged data even more aggressively. We firstly conducted experiments on 16 nodes with 8 bits as before. According to Figure 4(a), Alg. 1 and Alg. 2 on 16 nodes can still achieve basically same convergence rate as allreduce, which shows the scalability of our algorithms. However, they can not be comparable to allreduce with 4 bits,

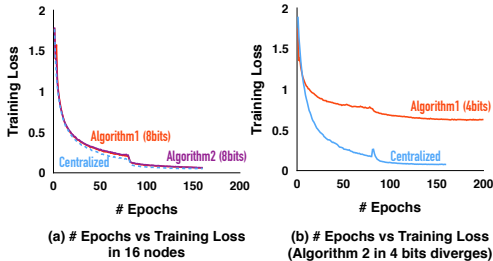

Figure 4: Comparison of Alg. 1 and Alg. 2

as is shown in 4(b). What is noteworthy is that these two compression approaches have quite different behaviors in 4 bits. For Alg. 1, although it converges much slower than allreduce, its training loss keeps reducing. However, Alg. 2 just diverges in the beginning of training. This observation is consistent with our theoretical analysis.

## 6 Conclusion

In this paper, we studied the problem of combining two tricks of training distributed stochastic gradient descent under imperfect network conditions: quantization and decentralization. We developed two novel algorithms or quantized, decentralized training, analyze the theoretical property of both algorithms, and empirically study their performance in a various settings of network conditions. We found that when the underlying communication networks has *both* high latency and low bandwidth, quantized, decentralized algorithm outperforms other strategies significantly.

**Acknowledgments**

Hanlin Tang and Ji Liu are in part supported by NSF CCF1718513, IBM faculty award, and NEC fellowship. CZ and the DS3Lab gratefully acknowledge the support from Mercedes-Benz Research & Development North America, Oracle Labs, Swisscom, Zurich Insurance, Chinese Scholarship Council, and the Department of Computer Science at ETH Zurich.

## Footnotes

[1]A real number is randomly quantized into one of closest thresholds, for example, givens the thresholds $\{0, 0.3, 0.8, 1\}$, the number "0.5" will be quantized to 0.3 with probability 40% and to 0.8 with probability 60%. Here, we assume that all numbers have been normalized into the range $[0, 1]$.

[2]A real number $z$ is set to 0 with probability $1 - p$ and to $z/p$ with probability $p$.

[3]Actually each neighbor of node $j$ maintains a replica of $\boldsymbol{x}_t^{(j)}$.

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
