[Reviews · NeurIPS 2018]

Reviewer 1



This paper deals with combining quantization with decentralized SGD in order to design algorithms which work well in networks with both high latency and low bandwidth. The authors propose two extensions of an existing decentralized SGD algorithm where the model updates are quantized, and analyze their convergence guarantees. Some experiments illustrate the approaches. The problem tackled in this paper, namely how to combine decentralization and quantization, seems interesting from a theoretical viewpoint. This is motivated by learning on networks with both high latency and low bandwidth. Can the authors give some concrete examples of machine learning problems/use-cases where one has these conditions combined? The effect of quantization is studied through high-level assumptions on the schemes, such as unbiasedness and independence across nodes and time (Assumption 1.6), low variance (Assumption 2), and a quantity related to a signal-to-noise ratio. This has the merit of generality, as it abstracts away the details of the particular quantization scheme. The authors present two algorithms to achieve the same goal. My main concern about this paper is that I do not see any argument in favor of Algorithm 1 compared to Algorithm 2. As stated by the authors, the convergence of Algorithm 1 is conditioned upon a hard assumption on the quantization strength, suggesting that it diverges if the quantization is too aggressive (and this is actually confirmed in the experiments). On the other hand, the convergence rate of Algorithm 2 smoothly degrades with the quantization strength, which is natural (the quantization noise should eventually dominate) and much nicer. Again this is confirmed numerically. In the end, all results (theoretical and empirical) show that Algorithm 2 is better, which makes Algorithm 1 useless. This raises the question of whether Algorithm 1 should be presented at all. I am also not fully convinced by the experiments: - The authors skip the comparison to a centralized but quantized algorithm, which they justify by the fact that "the difference between decentralized 8bits and centralized 8bits would be similar to the original decentralized training paper". However the centralized version may withstand quantization better as it averages more updates, which would make the proposed algorithms less appealing. - Experiments with a network of only 8 nodes is a bit disappointing, as an important motivation for decentralized algorithms is to scale to large networks. Having both low bandwidth and high latency on such a small network may also seem unlikely. It would be much more convincing if the authors provided evidence that their algorithm still performs well on a much larger network (this can be simulated if needed). Other comments: - Algorithms 1-2 should probably have a for loop over the workers - What is the quantization scheme used in the experiments? - In Section 5, there are many occurrences of the term "Allblackuce" (which I guess refers to Allreduce)

Reviewer 2



The paper presents two approaches to combining decentralized learning with model compression. Both method use gossip communication over a given network of nodes and model averaging. Before sharing the models with other nodes, they are compressed by either by using an unbiased compression on the difference of the model to the last version (the algorithm is called DCD-PSGD), or by an unbiased compression of a linear extrapolation of the model using the last version (ECD-PSGD). For both approaches, their convergence rate is theoretically analyzed. The combination of decentralized learning and model compression allows to handle networks with both low bandwidth and high latency. This claim is substantiated by an empirical evaluation. Quality: The idea of using compression in decentralized learning seems natural but interesting and the results appear to be useful. The technical content is well-presented and sound, the proofs appear to be correct. The claims of the paper are supported by both the theoretical analysis and the experiments. There is one minor issue I have. In Section 4.1 the paper claims a linear speedup of the DCD-PSGD algorithm (and the same for ECD-PSGD in Section 4.2). This definitely holds as long as the computational cost of calculating the average is in O(1), which is true as long as the network has an upper bounded node degree. However, for a fully connected network, the cost for calculating the average at each node would be in O(n), which is a constant overhead. This should reduce the speedup to sublinear. If this is correct, I would ask the authors to add an assumption, like an upper bounded node degree of the network, where the bound does not depend on the network size. Clarity: The paper is very well-written and clearly structured. The theoretical parts are comprehensible. The paper is overall a pleasure to read. There are some minor issues with English style and grammar, so I would ask the authors to have the paper proof-read. There are also some minor points and typos: - In Section 4.1 "Consistence with D-PSGD": I guess, ECD-PSGD should be DCD-PSGD, right? - In Section 4.2, Eq. (3) there is a t missing before x_t^(j) - In Section 5: it would be helpful to know which exact compression operator was used - In Section 5.4: blackucing should be reducing - In Appendix A.2, proof of Lemma 7: in the equation after "Therefore it yields", I don't see a difference between the lines 3 and 4. I guess one is a copy artifact. I would also like to ask the authors to check the formatting of the paper, as it seems to be a bit non-standard: the abstract is in italic, the placement of Figure 1 and the Definitions and Notations subsection is unusual, and there are weird line spaces (e.g., 226 to 227). Originality: Both decentralized learning by model averaging and communication compression have been studied before, as the authors admit, but their combination is relatively new (it was discussed in an arxiv paper of Konecny et al. "Federated Learning: Strategies for Improving Communication Efficiency" that is not yet published, but there it is only studied empirically). Moreover, they are to the best of my knowledge the first to theoretically analyse this approach. Thus, the main contribution of the paper are two novel and theoretically sound algorithms for decentralized learning with compression. Significance: Even though not groundbreaking, I really liked the paper. The algorithms it proposes seem to be useful for decentralized machine learning in many real-world application. The algorithms are proven to be theoretically sound and the empirical evaluation indicates the usefulness of the approach. The author's response has addressed my concern about the assumption of O(1) runtime complexity for calculating the average.

Reviewer 3



The authors investigate the convergence properties of two decentralized stochastic gradient algorithms that implement quantizations as a means of compressing communication. The algorithms fit into the main framework of decentralized first-order methods for consensus problems originally proposed in A. Nedic and A. Ozdaglar in 2009, with the results established in this paper for non-convex problems following more closely the work in X. Lian et al. 2017a and X. Lian et al. 2017b. The paper is in general well-written, although in some parts it lacks the required details that are needed to fully follow the presentation. For instance, one of the main points of the paper is to focus on notions of latency and bandwidth, but these are never defined, nor described (the reader is only referred to other literature, but this does not seem sufficient given the emphasis of the paper). In this sense, the paper falls short of providing the tools that are needed to appreciate the contribution of this work. For instance, can the authors' theory describe the plots in figure 8 when different latencies are considered? The literature review is by no means exhaustive of the variety and depth of work done in decentralized learning, but at least encompasses some of the main contributions in the field. The math and the formalism seem sound, though not always exhaustive (for instance, the functions F_i's in (1) are not described). The convergence results that are established in Section 4.1 and Section 4.2 are possibly misleading, as they treat the spectral gap of the communication matrix as a constant. On the other hand, it is well known that this quantity depends on the number of workers (n) in different ways according to the network topology being investigated (see J. Duchi, A. Agarwal, M. Wainwright, "Dual Averaging for Distributed Optimization: Convergence Analysis and Network Scaling" IEEE 2012, for instance). In fact, the numerical simulations presented in Section 5 consider the ring topology, which is one of the extreme examples in terms of how badly the spectral gap depends on the number of workers (quadratically). For instance, how would the result of Corollary 2 change when the dependence of the spectral gap with $n$ is taken into account?